# Alternaria Brown Spot Alters the Bacteriome with *Alternaria*–Bacteria Interactions in Mature Citrus Fruits

**DOI:** 10.3390/jof11110778

**Published:** 2025-10-28

**Authors:** Quan Chen, Wenbin Kong, Jinhui He, Xianwen Zhou, Yuan Huang, Zhongxian Liu, Feng Huang

**Affiliations:** 1Chongqing Three Gorges Academy of Agricultural Sciences, Wanzhou 404155, China; chenquan0616@126.com (Q.C.);; 2Plant Protection Research Institute, Guangdong Academy of Agricultural Sciences, Key Laboratory of Green Prevention and Control on Fruits and Vegetables in South China Ministry of Agriculture and Rural Affairs, Guangdong Provincial Key Laboratory of High Technology for Plant Protection, Guangzhou 510640, China; 3Chongqing Municipal Agricultural Technology Promotion Station, Liangjiang 401120, China; 4Economic Crop Development Center in Wanzhou District of Chongqing City, Wanzhou 404155, China; 5Chongqing Kaizhou District Agricultural Development Service Center, Kaizhou 405400, China

**Keywords:** Alternaria brown spot, citrus, Hongjv, microbial community, *Sphingomonas*

## Abstract

Alternaria brown spot is an important fungal disease in citrus. The infection of young citrus organs usually coincides with strong rainfall, which causes low efficiency of fungicides and the outbreak of this disease. Here, the microbiomes of the asymptomatic peels, the spot edge, and the center of citrus fruits were compared to reveal the commensal microbes as alternative control methods for the *Alternaria* pathogen. As the disease severity increased from the asymptomatic peels to the spot edge and the center, the bacterial communities were more severely changed than the fungal communities. Both the bacterial diversity, represented by the Shannon diversity index, and the bacterial composition and structure significantly decreased and altered, respectively. Increased *Alternaria*, in relative abundance, correlated positively with bacterial genera like *Massilia* and *Sphingomonas*, while negatively correlating with bacterial genera like *Delftia* and *Pantoea*. In addition, *Alternaria* fASV1 positively correlated with several top ASVs of 1174_901_12 and *Sphingomonas*. These results suggest that the bacterial communities respond to Alternaria brown spot by *Alternaria*–bacteria cross-kingdom interactions; these responsive bacteria are worth testing experimentally.

## 1. Introduction

*Citrus*, as a plant genus, includes a lot of fruit and medicinal plant species [1]. Many cultivated citrus varieties, such as those of sweet oranges, mandarins, and pomelos, are well known and popular around the world [1,2]. During the production of citrus fruits, diseases caused by microbes are a huge threat from sprout to fruit storage [3]. For example, the fungal diseases Alternaria brown spot, melanose, anthracnose, and greasy spot, caused by phylogenetically diverse fungal species, are commonly seen in citrus production areas in China [3].

Alternaria brown spot, mainly caused by the *Alternaria alternata* tangerine pathotype, is widely distributed in many citrus-cultivating countries such as USA, South Africa, Colombia, Turkey, Spain, Brazil, and Argentina [4,5]. In China, it was formally reported in the year 2010 [6]. Since then, this disease has been found to be a major fungal disease on different citrus varieties across many citrus production areas in China [7,8]. For example, the Hongjv in Chongqing Municipality, the Ougan in Zhejiang Province, the Ponkan in Zhejiang and Hunan Province, and the Gonggan in Guangdong Province have been reported to have Alternaria brown spot as a major fungal disease [8]. Alternaria brown spot usually occurs on young citrus leaves, twigs, and fruits; the infected organs will soon drop off and rot in the orchard [5]. However, some infected fruits can still hang on the tree until they mature; this allows long-term interactions between the pathogen and the plant microbiome.

The plant microbiome has been reported as an important player in plant–pathogen interaction [9,10]. For example, the below and aboveground organs of chili pepper (*Capsicum annuum* L.) could recruit beneficial bacteria against the fungal pathogen of *Fusarium* wilt disease [9], and the leaf of citrus could also enrich beneficial bacteria against the fungal pathogen *Diaporthe citri* after infection [10]. These studies imply that the plant microbiome is abundant with potential biocontrol agents, which can be isolated, inoculated, and commercially developed to control plant diseases [11]. Alternaria brown spot is a tough-to-control plant disease; its successful suppression heavily relies on intense fungicide application and less rainfall during the growth of young citrus organs [12,13]. However, the period of the growth of young citrus organs coincides with frequent heavy rainfall in most citrus production areas in China, which may cause the outbreak of Alternaria brown spot once in several years. Based on this, it is meaningful to study the response patterns of the citrus microbiome to Alternaria brown spot, which may offer new methods for the control of this citrus disease.

In this study, mature Hongjv fruits, with or without Alternaria brown spot, were collected in an orchard in Chongqing Municipality. The fruit peels were dissected into different tissues according to the necrosis spot, namely, spot center, edge, and asymptomatic peel. We aimed to compare the following: (1) the differences in the fungal and bacterial richness, diversity, composition, and structure; (2) the distribution of pathogen-associated *Alternaria* and major fungal and bacterial taxa; and (3) the fungi–bacteria interactions, especially *Alternaria*–bacteria interactions among the different spot tissues.

## 2. Material and Methods

### 2.1. Fruit Sampling

The fruits were sampled on 25 December 2024 in an orchard in Xiaozhou Village, Xiaozhou Town, Wanzhou District, Chongqing City. The variety of the citrus trees was Hongjv (*Citrus reticulata* cv. *Hongjv*), which is an ancient and local citrus variety. Hongjv is sensitive to Alternaria brown spot; it has been found to be infected with this disease from 2010 [14]. During the sampling, six trees were selected, and six to ten healthy and diseased mature fruits were picked from each tree, respectively. The brown spot of fruits from the same tree was cut out and mixed as a sample for the spot center (Center); the diameter of the spot center was about 0.5 cm in average (Figure 1). From the spot edge, a concentric circle with another 0.5 cm in diameter was cut out and mixed as a sample for the spot edge (Edge). The control fruit peels were cut out from asymptomatic fruits (Asym) using the same collection method as that for the Center samples. All the collected peels were surface-sterilized by soaking in 2% bleach for 15 s and 70% ethanol for 15 s and rinsed in sterile water three times [15]. Then, all samples were stored under −80 °C.

### 2.2. DNA Extraction and Amplicon Sequencing

Microbial genomic DNA was extracted from the ground peels using the CTAB (cetyltrimethylammonium bromide) method [16]. Partial nucleotide sequences were amplified by polymerase chain reaction (PCR) with the fungal primer pair ITS1F (5′-CTTGGTCATTTAGAGGAAGTAA-3′)/ITS2-2043R (5′-GCTGCGTTCTTCATCGATGC-3′) of fungal nuclear ribosomal internal transcribed spacer (ITS rDNA) and the bacteria primer pair V3F (5′-ACTCCTACGGGAGGCAGCA-3′)/806R (5′-GGACTACHVGGGTWTCTAAT-3′) of bacterial 16S rRNA V3 + V4 region, respectively [17,18]. The PCR system followed the protocol of the Phusion High-Fidelity PCR Master Mix (New England Biolabs Inc., Ipswich, MA, USA). The PCR cycle was designed as follows: initially 94 °C for 5 min, 35 cycles of 94 °C for 45 s, 56 °C for 30 s, 72 °C for 30 s, final extension at 72 °C for 10 min. PCR was conducted in triplicate for each sample and then mixed as an amplicon. The amplicon was verified on 2% agarose electrophoresis gel and extracted again with a Qiagen Gel Extraction Kit (Qiagen, Germany). Sequencing libraries were prepared using TruSeq^®^ DNA PCR-Free Library Preparation Kit (Illumina, San Diego, CA, USA) and verified on the Qubit@ 2.0 Fluorometer (Thermo Scientific, Santa Clara, CA, USA) and Agilent Bioanalyzer 2100 system (Agilent Technologies, Santa Clara, CA, USA). The libraries were sequenced on the Illumina HiSeq2500 platform (Illumina, San Diego, CA, USA) by BioMarker Co., Ltd. (BioMarker, Beijing, China).

### 2.3. Bioinformatics Analysis

After sequencing, 6,961,438 fungal and 5,298,017 bacterial paired-end raw reads were generated from 15 samples (the other tree samples of Asym were discarded because of low DNA quality). The raw reads were filtered by Trimmomatic version 0.39 [19] and cutadapt v1.9.1 [20] to remove the following: (1) short sequences (<200 bp), adapters, primers, poly-bases and (2) low-quality reads with more than 10% unknown nucleotides and less than 80% Q-value > 20 bases. After that, 4,411,222 (294,081 on average) fungal and 4,906,153 (327,077) bacterial clean reads were assembled using FLASH (v1.2.7, http://ccb.jhu.edu/software/FLASH/ , accessed on 4 March 2025). De-noising, removal of chimeric sequences, and amplicon sequence variant (ASV) clustering were all performed using dada2 [21] in QIIME2 2020.6 [22]. And then, a total of 12,173 (812) fungal and 36,854 (2457) bacterial ASVs were annotated using a Bayesian classifier against the UNITE (https://unite.ut.ee/ , accessed on 5 March 2025) and SILVA138 (http://www.arb-silva.de/ , accessed on 5 March 2025) databases, respectively.

### 2.4. Statistical Analysis

The workflow of statistical analysis was used for both the fungal and the bacterial communities. The ASV table was generated in QIIME2 2020.6 [22] and then normalized to the sample with the lowest sequence number. All the available alpha diversity indices (observed species, Shannon, Simpson, ACE, Chao1, Good’s coverage, and phylogenetic diversity) of each sample and the relative abundances of each taxon, from ASVs to microbial Kingdom, were calculated in the same platform of QIIME2. However, only the microbial observed species, Shannon diversity index, and phylogenetic diversity were selected to reflect the microbial richness and α diversity. PERMANOVA was used to compare the effects of spot tissues on the composition and structure of fungal and bacterial communities using the adonis command in the vegan package [23]. The Bray–Curtis distance of the ASV table among samples was subjected to principal coordinate analysis (PCoA) in the Ape package [24]; the first two PCoAs were scatter-plotted using the package ggplot2 [25]. The statistical comparison and test of significance were conducted in R software v4.0.0 (http://www.r-project.org/ , accessed on 26 May 2025). The comparisons among spot tissues were tested using the Kruskal–Wallis test followed by the Tukey post hoc pairwise test. The correlation analysis was performed and tested using the package Hmisc (https://hbiostat.org/R/Hmisc/ , accessed on 28 May 2025). Heatmap plotting and clustering were carried out in the package pheatmap [26]. The *p* values, smaller than 0.05, were accepted with significance.

## 3. Results

### 3.1. Fungal and Bacterial Diversity of the Fruits of Alternaria Brown Spot

The fungal and bacterial α diversities were represented by the number of observed species, Shannon diversity index, and phylogenetic diversity index (Figure 2). The mean number of fungal observed species (Figure 2A, not significant) was higher in the asymptomatic fruit peel (Asym, 1313) and in the spot edge (Edge, 1366.5) than in the spot center (Center, 1240.2). The Shannon diversity index was 6.5 and 5.7 in Asym and Edge, respectively, and also, on average, higher than 4.7 in the Center (Figure 2B, ns). The phylogenetic diversity index was 73.8 and 72.7 in Asym and Edge, respectively, and also, on average, higher than 64.9 in the Center (Figure 2C, ns). Similarly, for bacteria, the number of bacterial observed species was 2869 and 3096.7 in Asym and Edge, respectively, and higher than 2485.8 in the Center (Figure 2D, ns). The Shannon diversity index was 10.5 and 10 in Asym and Edge, respectively, and higher than 9.2 in the Center (Asym vs. Center, *p* < 0.05; Figure 2E). The phylogenetic diversity index was 12.6 and 13.6 in Asym and Edge, respectively, and also, on average, higher than 11.8 in the Center (Figure 2F, ns).

The β diversities of fungal and bacterial communities among Asym, Edge, and Center were compared by principal co-ordinate analysis (PCoA, Figure 3). The fungal communities of the Edge and Center were slightly deviated along PCoA1 (*p* = 0.079, ns), while the fungal communities of Asym could not be differentiated from those of Edge (Figure 3A). The bacterial communities of the Edge and Center were significantly differentiated by PCoA1 (*p* = 0.004); however, the bacterial communities of Asym could also not be differentiated from those of the Edge (Figure 3B).

### 3.2. Fungal and Bacterial Genera of the Fruits of Alternaria Brown Spot

At the genus level, the fungal communities of the mature fruit peel (Asym) were dominated by *Alternaria*, *Cladosporium*, *Saccharomyces*, *Teunomyces*, *Zasmidium*, *Fusarium*, *Cyberlindnera*, *Colletotrichum*, *Ramularia*, and *Aureobasidium*, which accounted for 37.4% of the relative abundance of the whole community (Figure 4A). The bacterial communities were dominated by *Methylobacterium*_*Methylorubrum*, *Sphingomonas*, *Erwinia*, 1174_901_12, *Pseudomonas*, *Exiguobacterium*, *Weissella*, *Gemmobacter*, *Ligilactobacillus*, and *Romboutsia*, which accounted for 15.4% of the relative abundance of the whole community (Figure 4B).

In response to Alternaria brown spot, the relative abundances of the fungal genera *Alternaria* (16.6% and 29.2%, to 5%; *p* < 0.05), *Cladosporium* (14.5 and 12, to 8.3), *Colletrotrichum* (2.6 and 2.7, to 0.7) and the bacterial genera *Methylobacterium*_*Methylorubrum* (6.9 and 15.7, to 1.1), *Sphingomonas* (3.2 and 7.4, to 1.1; *p* < 0.05), *Erwinia* (1.6 and 8, to 0), 1174_901_12 (1 and 8, to 0.3) increased in the spot Edge and Center compared with those in the Asym. The relative abundances of the fungal genera *Teunomyces* (1, to 3.4 and 4.2; *p* < 0.05), *Zasmidium* (1.3, to 2.5 and 3.8), *Fusarium* (2, to 2.9 and 2.1) and the bacterial genera *Gemmobacter* (0.5, to 1.6 and 2.7; *p* < 0.05), *Ligilactobacillus* (0.9, to 2.1 and 1.8; *p* < 0.05), *Romboutsia* (0.9, to 1.6 and 2.2; *p* < 0.01) decreased in the spot Center compared with those in the Asym and Edge. The comparison results of other fungal and bacterial genera are listed in Appendix A.

### 3.3. Fungi–Bacteria Correlation Pattern of the Fruits in Alternaria Brown Spot

The top 50 fungal and the top 50 bacterial genera, ranked by their means of relative abundance, were selected for cross-kingdom correlation analysis. And then, the correlation coefficients between fungal and bacterial genera were plotted and clustered in a heatmap (Figure 5). The fungal and bacterial genera were both divided into two clades, namely F1/F2 and B1/B2, respectively. The fungal clade F1, consisted of 20 genera, mainly correlated positively with the bacterial clade B1, consisting of 7 genera, with 30 pairs of significantly positive correlations (red with *, *p* < 0.05). For example, the fungal genus *Alternaria* correlated positively with the bacterial genera *Massilia* (*R*^2^ = 0.59, *p* < 0.05) and *Sphingomonas* (*R*^2^ = 0.57, *p* < 0.05). The fungal clade F2, consisted of 30 genera, also mainly correlated positively with the bacterial clade B2, consisting of 43 genera, with 299 pairs of significantly positive correlations (red with *, *p* < 0.05). On the contrary, the fungal clade F1 mainly correlated negatively with the bacterial clade B2 with 41 pairs of significantly negative correlations (blue with *, *p* < 0.05) and one pair of significantly positive correlation (*Didymella* to *Paenisporosarcina*, *R*^2^ = 0.53, *p* < 0.05). For example, the fungal genus *Alternaria* correlated negatively with the bacterial genera *Delftia* (*R*^2^ = −0.52, *p* < 0.05) and *Pantoea* (*R*^2^ = −0.53, *p* < 0.05). The fungal clade F2 mainly correlated negatively with the bacterial clade B1 with 20 pairs of significantly negative correlations (blue with *, *p* < 0.05).

The relative abundance of *Alternaria* was mainly represented by a single fASV1, whose mean relative abundances were 4.9%, 16.5%, and 29.2% in the fungal communities of the Asym, spot Edge, and Center, respectively (Asym vs. Center, *p* < 0.05; Figure 6A). fASV1 consistently and positively correlated with the major ASVs of two bacterial genera 1174_901_12 and *Sphingomonas* of Alphaproteobacteria (Figure 6B–H). Specifically, fASV1 significantly correlated with bASV8 (*R*^2^ = 0.27, *p* < 0.05; Figure 6B), bASV19 (*R*^2^ = 0.3, *p* < 0.05; Figure 6C), bASV22 (*R*^2^ = 0.33, *p* < 0.05; Figure 6D), and bASV49 (*R*^2^ = 0.29, *p* < 0.05; Figure 6E) of 1174_901_12 and also with bASV30 (*R*^2^ = 0.64, *p* < 0.01; Figure 6F), bASV33 (*R*^2^ = 0.32, *p* < 0.05; Figure 6G), and bASV42 (*R*^2^ = 0.45, *p* < 0.01; Figure 6H) of *Sphingomonas*. Even though the bacterial genus *Methylobacterium_Methylorubrum* had the highest relative abundance in the bacterial communities of the spot Edge and Center, fASV1 did not correlate significantly with its major ASVs, such as bASV7 (Figure 6I) and bASV13 (Figure 6J). On the contrary, fASV1 negatively correlated with bASV106 of *Delftia* (*R*^2^ = −0.4, *p* < 0.01; Figure 6K) and bASV57 of *Escherichia_Shigella* (*R*^2^ = −0.17; Figure 6L).

## 4. Discussion

Hongjv is a local tangerine variety that is sensitive to Alternaria brown spot [14,27]. Even with effective fungicides and reasonable fungicide application methods, this disease is still prevalent in the village in which we first identified it in the year 2010 [14]. This is because there has not been a once-and-for-all method to completely eradicate this pathogen for the last fifteen years. The different sampled citrus fruit peel tissues, including the asymptomatic fruit peel, spot edge, and center, roughly reflected the severity of pathogen infection. This could be confirmed by the relative abundance of *Alternaria* fASV1 in different peel tissues, which increased from 4.9% to 16.5% to 29.2% in the fungal communities of the asymptomatic fruit peel, spot edge, and center (Figure 6A). Interestingly, *Alternaria* fASV1 existed as one of the dominating fungal ASVs even in the peels of healthy fruits, which implies its lifestyle as commensal endophyte or latent pathogen [28,29]. Previous studies have found different pathotypes and phylotypes of *Alternaria alternata* on citrus, they were unable to differentiate using taxonomical barcoding genes [8,29,30]. Accordingly, it is hard to prove that all fungal strains, assigned to fASV1, are the tangerine pathotypes of *Alternaria alternata* with the current knowledge and methods. However, fASV1 drastically and significantly increased in the infected spot edge and center, which confirms that fASV1 contains the main tangerine pathotypes of *Alternaria alternata*. The pathotypes of fASV1 were prevalent in the spot edge and center, changing the nutritional state of fruit peels by inducing citrus cell necrosis.

Furthermore, the increasing severity of pathogen infection may confer an increasing survival pressure on the microbiome of citrus peels [9]. From Figure 2, the fungal and bacterial richness and diversity tended to decrease from the asymptomatic fruit peel to the spot edge to the spot center, which negatively coincided with the relative abundance of fASV1. In other studies, it was also observed that infection with the pathogen decreased the richness and diversity of the plant microbiome [31,32]. This may be because the pathogen infection increases the competitiveness of other opportunistic pathogens [31]) and beneficial microbes [32], thus restricting the growth of most microbes without obvious competing advantages.

The altered composition and structure of the microbial communities, from the asymptomatic fruit peel to the spot edge to the spot center, also confirms the increased competition among the peel fungi and bacteria. From Figure 3, the bacterial communities were more sensitive to infection by the fungal pathogen than the fungal communities were, which suggests that the fungi–bacteria inter-kingdom interactions are more prevalent and influential than the respective intra-kingdom interactions [33,34].

Under the influences of the fungi–bacteria inter-kingdom interactions, several major bacterial genera, such as *Methylobacterium*_*Methylorubrum*, *Sphingomonas*, *Erwinia*, and 1174_901_12, increased in relative abundances along with the increased severity of pathogen infection. Of which, the whole genera or the major ASVs of *Sphingomonas* and 1174_901_12 were detected with significant correlation with the genus *Alternaria* or the *Alternaria* fASV1. In plants, bacteria of the genus *Sphingomonas* are closely involved in improving plant health and resistance [35,36]. For example, the *Sphingomonas* species were enriched in relative abundance and with genes of iron complex outer membrane receptors in *Diaporthe citri*-infected citrus leaves; one representative *Sphingomonas* isolate was further proven to have strong suppression ability against *Diaporthe citri* in iron-deficient conditions [10]. This implies that the positive correlations between the *Sphingomonas* ASVs and *Alternaria* fASV1 are worth testing in experiments.

## 5. Conclusions

The occurrence of Alternaria brown spot significantly affected the bacterial communities of mature citrus fruits, while it only mildly affected the fungal communities in some taxa. The bacterial richness and diversity were reduced and the composition and structure of bacterial communities were altered, which were associated with the positive interactions between bacterial major taxa, such as 1174_901_12 and *Sphingomonas*, and *Alternaria* fASV1. These results suggest that infection with the *Alternaria* pathogen works as a selection factor; it restricts the bacteria to live in the necrotic citrus tissues by intensifying the cross-kingdom interactions. Our study is meaningful for understanding the microbial interactions triggered by Alternaria brown spot and for determining bacterial resources for disease control. The bacterial ASVs of 1174_901_12 and *Sphingomonas* need to be isolated and further used to test their co-occurrence with *Alternaria alternata*.

## Figures and Tables

**Figure 1 jof-11-00778-f001:**
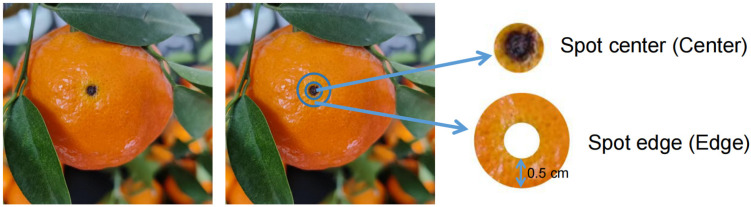
Schematic diagram of the sampling of spot edge and center of Alternaria brown spot from citrus fruit peel.

**Figure 2 jof-11-00778-f002:**
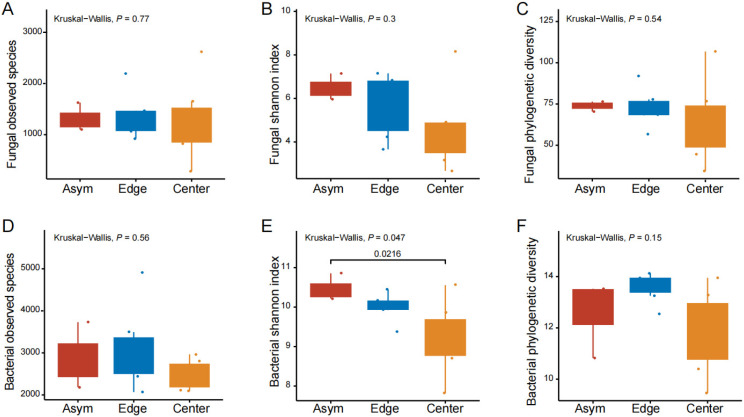
Fungal and bacterial observed species, Shannon diversity index, and phylogenetic diversity among asymptomatic peels, spot edge, and center. (**A**–**C**) Fungal communities; (**D**–**F**) bacterial communities.

**Figure 3 jof-11-00778-f003:**
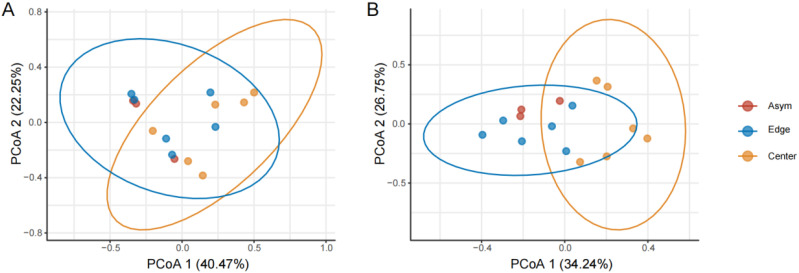
Scatter plot of the first two principal co-ordinates of the fungal and bacterial communities of asymptomatic peels, spot edge, and center. (**A**) Fungal communities; (**B**) bacterial communities. The principal co-ordinate analysis was based on microbial ASVs; the oval lines were drawn based on 95% confidence intervals.

**Figure 4 jof-11-00778-f004:**
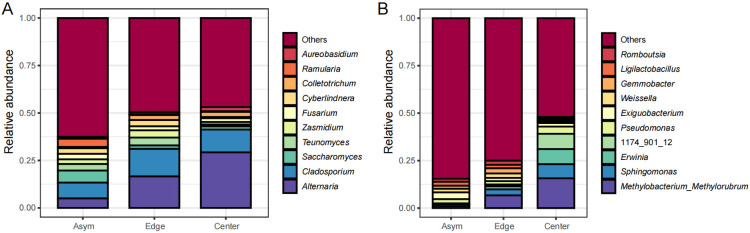
Top 10 fungal and bacterial genera ranked in relative abundance of asymptomatic peels, spot edge, and center. (**A**) Fungal communities; (**B**) bacterial communities.

**Figure 5 jof-11-00778-f005:**
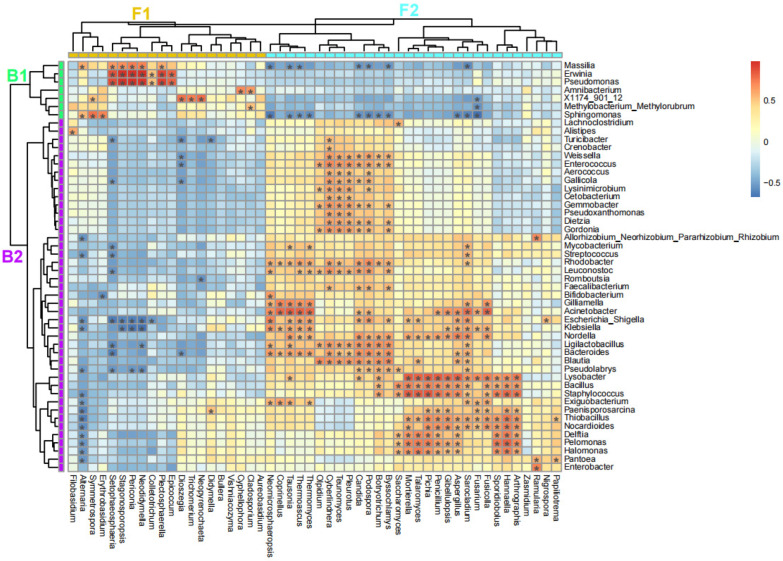
Heatmap plot of the correlation coefficients between the top 50 fungal and bacterial genera. F1 and F2, fungal clades 1 and 2; B1 and B2, bacterial clades 1 and 2. Color of grid represented by the value of correlation coefficients; “*” denotes *p* value < 0.05.

**Figure 6 jof-11-00778-f006:**
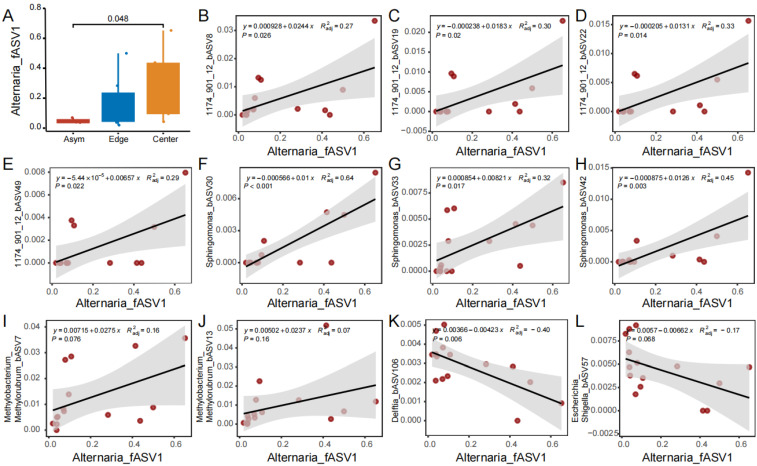
Correlation of bacterial ASVs with *Alternaria* fASV1. (**A**) The relative abundance of *Alternaria* fASV1 in asymptomatic peels, spot edge, and center; (**B**–**E**) correlation of ASVs of 1174_901_12 with fASV1; (**F**–**H**) correlation of ASVs of *Sphingomonas* with fASV1; (**I**,**J**) correlation of ASVs of *Methylobacterium_Methylorubrum* with fASV1; (**K**) correlation of *Delftia* bASV106 with fASV1; (**L**) correlation of *Escherichia_Shigella* bASV57 with fASV1. The gray shadow along the regression line represents the 95% confidence interval.

## Data Availability

Raw sequence data of the amplicons, the codes in analyses, and plotting are available from the corresponding author (F.H.) upon reasonable request.

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
