# Peer review of "Alternaria Brown Spot Alters the Bacteriome with Alternaria–Bacteria Interactions in Mature Citrus Fruits"

_jof, 2025, doi:10.3390/jof11110778_

Round 1

Reviewer 1 Report

No major comments

3.1: Fungal and bacterial diversity of the fruits of Alternaria brown spot section: please add when the differences were or not significant in the text.

Figure 4. maybe in the “others” there is valuable information. Could the identification of the other bacterial and fungal genera be added in a supplementary material? Please write the bacterial ad fungal names in italics.

Line 264: replace has for was

Author Response

#reviewer 1

3.1: Fungal and bacterial diversity of the fruits of Alternaria brown spot section: please add when the differences were or not significant in the text.

Response 1: Done as suggested. Please check the part 3.1 again.

Figure 4. maybe in the “others” there is valuable information. Could the identification of the other bacterial and fungal genera be added in a supplementary material? Please write the bacterial ad fungal names in italics.

Response 2: Done as suggested. Thank you for this suggestion. We added two supplementary tables of these information, please check the Supple. Table 1 and 2.

Line 264: replace has for was

Response 3: Done as suggested.

Reviewer 2 Report

-

Alternaria Brown Spot Alters Bacteriome with Alternaria-Bacteria Interactions of Mature Citrus Fruits

The paper is thematically interesting and contributes to the understanding of the interactions between the Alternaria fungus and the bacterial community on citrus fruits, which has practical significance for the biological control of diseases.

Line 4 - Is this paper missing an author or is there a mistake?

However, there are certain shortcomings, which I will list in detail below:

INTRODUCTION

The Introduction section is concise and specific.

Lines 44-49 - It is clear to me that the authors focused on China when searching the scientific literature, but I believe that the disease is also present in some other countries. Please add information about the presence of the disease outside of China (example - references 24).

Lines 70 - why these two references? In my opinion, they are not necessary in this part where the aim of the paper is explained. The presence of these references in the first sentence of the Discussion section of this paper is justified.

MATERIALS AND METHODS

In several places in the text, the genus Alternaria is not written in italic. I ask the authors to correct this.

Line 82- I think this is a small sample - a small number of repetitions. It's not really relevant.

Line 115 - This is not a very representative sample.

RESULTS

The Results section is written correctly. The visual presentation is problematic (Figures 5 and 6). It seems quite confusing to follow the text and Figures 5 and 6. I suggest that the authors consider a clearer/more specific presentation if possible.

DISCUSSION

The Discussion section follows and adequately discusses the previous Results section.

The hypothesis of cross-kingdom interactions is not sufficiently experimentally confirmed correlation is not evidence of causation (experimental validation is needed).

Perhaps some more information should be added about the Sphingomonas genus and the potential role of bacteria.

CONCLUSION

State why these results are important. Also add what should be done in the future to improve this research.

GENERAL OPINION

Overall, the paper is well written. I find it interesting from a scientific perspective and have scientific potential, but several key points are questionable:

* Only six trees and a small number of biological replicates (15) are a weak statistical basis.

* The hypothesis of cross-kingdom interactions is not sufficiently experimentally confirmed correlation is not evidence of causation (experimental validation is needed).

* Figures 5 and 6 are confusing. I suggest considering a clearer presentation. It is very difficult to read the results.

If the authors accept the proposed comments and suggestions, I believe that this paper can be considered for publication.

Best regards,

Reviewer

Author Response

#reviewer 2

The paper is thematically interesting and contributes to the understanding of the interactions between the Alternaria fungus and the bacterial community on citrus fruits, which has practical significance for the biological control of diseases.

Response 1: Thank you for the comment on our manuscript.

Line 4 - Is this paper missing an author or is there a mistake?

Response 2: That was a typo error, we revised it. Thank you for reminding.

However, there are certain shortcomings, which I will list in detail below:

Response 3: We tried our best to solve all your suggestions, please check the manuscript again.

INTRODUCTION

The Introduction section is concise and specific.

Lines 44-49 - It is clear to me that the authors focused on China when searching the scientific literature, but I believe that the disease is also present in some other countries. Please add information about the presence of the disease outside of China (example - references 24).

Response 4: The reviewer is right, Alternaria brown spot is a worldwide fungal disease on citrus. We added a sentence to describe its occurrence in major citrus cultivating countries. Please check the second paragraph of the Introduction.

Lines 70 - why these two references? In my opinion, they are not necessary in this part where the aim of the paper is explained. The presence of these references in the first sentence of the Discussion section of this paper is justified.

Response 5: Agree. The two unnecessary references were deleted.

MATERIALS AND METHODS

In several places in the text, the genus Alternaria is not written in italic. I ask the authors to correct this.

Response 6: We check through the text including the Reference. Many “Alternaria” were changed to italic, only the “Alternaria” used along with “brown spot” was not in italic.

Line 82- I think this is a small sample - a small number of repetitions. It's not really relevant.

Response 7: Thank you for this question, this is a really important reminding of our work. However, each of our sample was a mix of 6-10 fruits, it was a pool of fruits from a tree. For example, the six samples of Center included about 36-60 diseased fruits. So, we hope the reviewer still think our sampling was reasonably designed.

Line 115 - This is not a very representative sample.

Response 8: The same to the Response 7.

RESULTS

The Results section is written correctly. The visual presentation is problematic (Figures 5 and 6). It seems quite confusing to follow the text and Figures 5 and 6. I suggest that the authors consider a clearer/more specific presentation if possible.

Response 9: We revised the part 3.3 where the Fig. 5 and 6 located. We hope that they are clear this time.

DISCUSSION

The Discussion section follows and adequately discusses the previous Results section.

Response 10: The Discussion is revised according to the following specific suggestions.

The hypothesis of cross-kingdom interactions is not sufficiently experimentally confirmed correlation is not evidence of causation (experimental validation is needed).

Response 11: The reviewer is right. We are also very interested in the Alternaria-bacteria interactions, especially our finding of the fASV1-Sphingomonas correlation. However, it’s a little regretful that we can not fulfill this test in a very short time, especially considering our limited students, workers, and funding.

Perhaps some more information should be added about the Sphingomonas genus and the potential role of bacteria.

Response 12: Done as suggested. We added an example of the functions of Sphingomonas against another fungal pathogen of citrus.

CONCLUSION

State why these results are important. Also add what should be done in the future to improve this research.

Response 13: Thank you for this reminding. Done as suggested.

GENERAL OPINION

Overall, the paper is well written. I find it interesting from a scientific perspective and have scientific potential, but several key points are questionable:

Response 14: Thank you again for your comments on our manuscript.

* Only six trees and a small number of biological replicates (15) are a weak statistical basis.

Response 15: We agree with the reviewer’ opinion. We should have collected more samples. The first author and the corresponding author are not from the same institution, we communicated online, so we did not make the sampling process very clear to each other. But as we explained in Response 7, our sample was a mix of sub-samples, namely, each sample consisted of 36 to 60 citrus fruits. In addition, we used strict adjust P value in our data analysis to avoid stochastic results. Based on these, we hope you think our results are publishable.

* The hypothesis of cross-kingdom interactions is not sufficiently experimentally confirmed correlation is not evidence of causation (experimental validation is needed).

Response 16: This is another drawback of our manuscript. Thank you again for this suggestion. We aim to test the findings from our data analysis, but it’s regretful we do not have many results at this time. Funding is not easy to get for the research on citrus fungal diseases, most of them are invested into citrus Huanglongbing.

* Figures 5 and 6 are confusing. I suggest considering a clearer presentation. It is very difficult to read the results.

Response 17: Figures 5 and 6 present the results of correlation analysis. We added some indicative words in the text of the part 3.3, and modified Fig. 5 by adding side bars, matching the colors, and improving the font size. We hope they are better to read this time.

If the authors accept the proposed comments and suggestions, I believe that this paper can be considered for publication.

Response 18: We really appreciate the suggestions by the reviewer, and tried our best to revise the manuscript according to all the suggestions. Two suggestions, on biological replicates and experimental validation, are hard for us to fulfill in a short time. There may be other methods to remedy these drawbacks of our research, we sincerely ask for another chance to revise the manuscript.